# The Influence of Environmental Polycyclic Aromatic Hydrocarbons (PAHs) Exposure on DNA Damage among School Children in Urban Traffic Area, Malaysia

**DOI:** 10.3390/ijerph19042193

**Published:** 2022-02-15

**Authors:** Nur Hazirah Hisamuddin, Juliana Jalaludin, Suhaili Abu Bakar, Mohd Talib Latif

**Affiliations:** 1Department of Environmental and Occupational Health, Faculty of Medicine and Health Sciences, Universiti Putra Malaysia (UPM), Serdang 43400, Malaysia; gs49787@student.upm.edu.my; 2Department of Biomedical Sciences, Faculty of Medicine and Health Sciences, Universiti Putra Malaysia (UPM), Serdang 43400, Malaysia; suhaili_ab@upm.edu.my; 3Department of Earth Sciences and Environment, Faculty of Science and Technology, Universiti Kebangsaan Malaysia, Bangi 43600, Malaysia; talib@ukm.edu.my

**Keywords:** polycyclic aromatic hydrocarbons (PAHs), children, urban traffic area, DNA damage

## Abstract

This study aimed to investigate the association between particulate PAHs exposure and DNA damage in Malaysian schoolchildren in heavy traffic (HT) and low traffic (LT) areas. PAH samples at eight schools were collected using a low volume sampler for 24 h and quantified using Gas Chromatography-Mass Spectrometry. Two hundred and twenty-eight buccal cells of children were assessed for DNA damage using Comet Assay. Monte-Carlo simulation was performed to determine incremental lifetime cancer risk (ILCR) and to check the uncertainty and sensitivity of the estimated risk. Total PAH concentrations in the schools in HT area were higher than LT area ranging from 4.4 to 5.76 ng m^−3^ and 1.36 to 3.79 ng m^−3^, respectively. The source diagnostic ratio showed that PAHs in the HT area is pyrogenic, mainly from diesel emission. The 95th percentile of the ILCR for children in HT and LT area were 2.80 × 10^−7^ and 1.43 × 10^−7^, respectively. The degree of DNA damage was significantly more severe in children in the HT group compared to LT group. This study shows that total indoor PAH exposure was the most significant factor that influenced the DNA damage among children. Further investigation of the relationship between PAH exposure and genomic integrity in children is required to shed additional light on potential health risks.

## 1. Introduction

Urbanisation and economic growth have been strongly associated with increased transportation demand and the number of road vehicles within cities [1,2]. Pollutants from traffic emission or known as traffic related air pollution (TRAP), were the major contributor to air pollution in Malaysia, particularly in urban areas. Particulate matter (PM) is considered one of the most harmful components of ambient pollutants to humans [3]. Recent studies in the city of Kuala Lumpur, Malaysia demonstrated that PM pollution in this area is greatly contributed by vehicle emissions [4,5,6,7,8].

Polycyclic aromatic hydrocarbons (PAHs) are one of the most important organic groups bound to particulate matter in terms of health risk. PAHs are ubiquitous, semi-volatile and persistent organic pollutants that are formed as by-products from the incomplete combustion of organic materials [9,10,11]. Due to their resistance to degradation processes, especially when bound to particles, they are transported over long distances and could be identified even in remote areas [12]. Several studies have shown that motor vehicle emissions contribute to PAHs pollution in urban areas [13,14,15,16,17] especially high-molecular weight particulate PAHs, which are primarily present in PM_2.5_ [18]. Several studies conducted in the Klang Valley also indicated the contribution of motor vehicle emissions to PAHs concentration [10,11,17,18,19].

The impact of PAHs on the environment and public health has sparked widespread concern due to their mutagenic and carcinogenic properties. Exposure to genotoxic substances such as PAHs can cause oxidative stress, which could lead to DNA damage and disturbances in DNA replication. Changes in DNA replication may cause mutation and lead to carcinogenic effects [20,21]. Evidence from previous studies suggests that environmental PAHs exposure may lead to increased DNA damage in children attending schools within 500 m of busy roads [22,23]. Children have a longer lifespan and are more likely to develop chronic diseases that can be triggered by early exposure. Chronic diseases and cancer caused by environmental toxicants are thought to evolve in phases that take years or even decades to develop from their initiation to clinical manifestation. Carcinogenic and toxic exposures in early childhood seems more likely to lead to disease than similar exposures later in life [24,25].

Previous studies that highlighted traffic emission as the primary contributor of PAHs in the Klang Valley [10,11,18,19] give the concern to assess the chronic health effects of inhalation PAHs among the susceptible population living in this area, especially children. In order to estimate the health risks from the exposure to PAHs, it is also essential to quantify the resultant changes that are effected through the application of biomarkers associated with the toxicants [23]. Educational buildings such as primary schools are among the most important buildings where children typically spend up to 1/3 of their time [26]. Therefore, understanding exposure to health concerned pollutants in these locations has become a priority for the scientific community [27,28,29,30]. To date, there is limited information on PAH exposure in school settings in Malaysia. Since children attending schools near congested urban areas are constantly exposed to air pollutants, this study is carried out to fill the knowledge gap on the association between PAHs from traffic emission and the chronic health effect concerning DNA damage among Malaysian children in the urban city of the Klang Valley.

## 2. Materials and Methods

### 2.1. Study Location

Klang Valley was selected as the location for this study because it is known as fast growing area and the most populated area in Malaysia. The cities in the Klang Valley region have a well-developed road network, with approximately 379,146 vehicles travelling around daily [31]. The school selection was based on proximity to high and low traffic areas. High traffic area is defined as areas within 500 m on either side of highways with an average daily traffic (ADT) volume of ≥18,000 vehicles, or within 100 m on either side of major roads with an ADT volume of ≥15,000 vehicles [32]. Children from primary schools in Kuala Lumpur and Gombak, were categorised as high traffic (HT) group. The schools in HT were designated H1, H2, H3 and H4. Meanwhile, low traffic is defined as area located 5 km away from nearby highways, major roads and industrial sites. The children of primary schools in Hulu Langat, were categorised as the low traffic (LT) group, with the schools on LT designated as L1, L2, L3, and L4. The map of study area is shown in Figure 1.

### 2.2. Study Population

This study includes a total of 228 school children studying at the selected primary schools in Klang Valley, specifically age between 7 to 11 years old who met the inclusion and exclusion criteria. Both genders, which are female and male, were selected among the Malay population to prevent gender bias of collected data. Only Malaysian citizens and Malay ethnicity were recruited to homogenize the samples. These criteria have been decided to control genetic differential factors, which could affect the final outcomes. Children who had a known history of medical problems were excluded in this study. This criterion has been decided because it has the potential to impair children’s physiological function and increase their susceptibility to pollutants [22]. Besides, children who had radiotherapy or chemotherapy in the previous 12 months or X-rays in the last three months were excluded. Any radiation exposure can affect DNA structure by inducing DNA strand breaks and thus affecting the significance of the results of this study [33]. Children who had a family history of cancer also were excluded. This criterion has been decided because he or she have higher risk of getting cancer [34].

### 2.3. Questionnaires

A total of 280 questionnaires and consent forms were distributed to eight schools and 228 respondents completed the questionnaire and gave consent for biological sampling, representing 81.4% of response rate. A validated questionnaire from American Thoracic Society (ATS-DLD-78-C) and International Study of Asthma and Allergies in Childhood (ISAAC) translated from English to Malay (Appendix A) were distributed to parents or guardians of children.

### 2.4. PM_2.5_ Sampling

Gravimetric sampling of PM_2.5_ were conducted using a low volume sampler, MiniVol Air Sampler (Airmetrics, Springfield, OR, USA; model 4.2). The filter paper used was 47 mm quartz microfiber filter papers (Whatman, Maidstone, Kent, UK; catalogue no. 1851-047). PAH samples were extracted from the filter paper. MiniVol Air Sampler was set up on the ground with its stand to sample air at a rate of 5 L/min, indoor and outdoor for 24 h. The air sampler was located approximately 1.0 m above the floor and placed at the back of the selected classroom. For outdoor monitoring, the sampling was conducted in a safe area, which is near the main entrance gate of the school or guard posts. A total of 64 samples and 8 blank samples (one for each school) were collected during the sampling campaign.

### 2.5. PAHs Extraction

The filter paper was cut into small pieces (approximately 1 cm × 1 cm) in a 50 mL glass bottle. 0.1 mL of 1.5 ppm PAHs surrogate internal standards; anthracene-d_10_, p-terphenyl-d_14_, benz[a]anthracene-d_12_, and perylene-d_14_ (SUPELCO, St. Louis, MO, USA) were spiked into all samples for recovery assessment. 10 mL of dichloromethane (DCM) and n-hexane with ratio 5 mL:5 mL was added into the glass bottle. Next, the mixture was sonicated using a bath sonicator (Elma, Singen, Germany) for 30 min (2 min run and 1 min rest × 15 cycles). This procedure was repeated three times and the extracts were combined. Then, the extract was concentrated to approximately 0.2 mL under a gentle blow of nitrogen gas (N_2_). Silica SPE cartridges (LiChrolut RP-18 1000 mg 6 mL, Merck, Germany; catalogue no. 1021220001) were used for clean-up process and pre-concentration of samples. The final step of SPE was elution by DCM:n-hexane (3.5 mL:6.5 mL). The eluent was further reduced to 0.1 mL via a gentle stream of N_2_ gas before transferring to a 2 mL autosampler vial containing a vial insert.

The PAHs were quantified using gas chromatography–mass spectrometry (GC-MS) instrument (Agilent Technologies, Santa Clara, CA, USA; model 6890N/5975), fitted with HP-5MS capillary column (30 m × 250 µm). Helium was used as a carrier gas with a flow rate of 1.0 mL/min. The sample was injected at 200 °C using the splitless mode. The temperature of the GC column was programmed as follows: initial 40 °C, followed by a temperature increase to 150 °C (8 °C per min), and an increment of 4 °C per min to 310 °C for a 6 min hold. Mass spectrometry was acquired using the electron ionization (E.I.) mode.

The concentrations of 16 United States Environmental Protection Agency (US EPA) priority PAHs were determined in this study including 3-rings PAHs; acenaphthene (ACP), acenaphthylene (ACY), anthracene (ANT), fluorene (FLR), phenanthrene (PHE), 4-rings PAHs; fluoranthene (FLT), pyrene (PYR), benzo(a)anthracene (BaA), chrysene (CYR), 5-rings PAHs; benzo(k)fluoranthene (BkF), Benzo[b]fluoranthene (BbF), benzo(a)pyrene (BaP), dibenzo(a,h)anthracene (DhA) and 6-rings PAHs; indeno(1,2,3-cd)pyrene (IcP) and benzo(ghi)perylene (BgP); except for 2-rings PAHs, naphthalene (NAP) owing to its high volatilities. The GC-MS instrument was calibrated with standard mixtures of PAHs (PAH Calibration Mix, SUPELCO, USA; catalogue no. SA_CRM47940). Five points (0.2, 0.5, 1.0, 1.5, 3.0 ppm) of standard PAHs mixtures were analysed for the establishment of the calibration curves. The correlation coefficients (R^2^) for linear regressions of the calibration curves were >0.99 in all cases. For every sample, procedural blanks were run to ensure that there were no significant background interferences. Blank filters were extracted and analysed using the same method with the actual samples. The limit of detection (LOD) of each PAHs were calculated based on five independent measurements of blank samples and its standard deviation. The recovery efficiency for the internal standards ranged from 79–113%. The LOD for individual PAHs compounds ranged from 0.01 to 0.17 ng m^−3^.

### 2.6. Health Risk Assessment

BaP-equivalent concentration (BaPeq), also known as toxicity equivalent concentration (TEQ), was used to evaluate the toxicity of PAHs. To calculate TEQ, the reference toxic equivalent factor (TEFs) of PAHs with respect to BaP were multiplied with the concentration of PAHs species [35] as shown on the following equation:(1)$$\begin{array}{c} \mathrm{TEQ}=0.001\;\left(\mathrm{ACY}+\mathrm{ACE}+\mathrm{FLR}+\mathrm{PHE}+\mathrm{FLT}+\mathrm{PYR}\right)+0.01\;\left(\mathrm{ANT}+\mathrm{BgP}+\mathrm{CYR}\right)\\ +0.1\left(\mathrm{BaA}+\mathrm{BkF}+\mathrm{BbF}+\mathrm{IND}\right)+\mathrm{BaP}+\mathrm{DhA} \end{array}$$

The carcinogenic risk of PAHs by respiratory exposure was estimated using the incremental lifetime cancer risk (ILCR) model [36,37]. The equation is as follows:(2)$$\mathrm{ILCR}=\frac{C\times \left(\mathrm{CSF}\times ∛\frac{\mathrm{BW}}{70}\right)\times \mathrm{IR}\times \mathrm{ED}\times \mathrm{EF}}{\mathrm{BW}\times \mathrm{AT}}\times \mathrm{cf}$$
where:

C (ng m^−3^)=TEQCSF (3.85 mg kg^−1^ day^−1^)=Inhalation cancer slope factor of BaPBW (kg)=Body weight (kg)IR (12 m^3^ day^−1^);=Inhalation rateED (6 years)=Exposure durationEF (250 day year^−1^)=Exposure frequencyAT (70 years × 365 days)=Averaging time of carcinogenic PAHs exposure [38]cf=Conversion factor

To reduce the uncertainties in risk assessment models, Monte-Carlo simulation using Crystal Ball software version 11.1.2.4 (Oracle Corp., Austin, TX, USA) was applied for probability and sensitivity analysis, where parameters were expressed by a series of data based on probability distribution. The results of sensitivity analysis were shown as rank correlation coefficients and the higher coefficient represents the most contributor in the uncertainty of calculated risk. Table 1 shows the variable distribution types applied in the Monte Carlo simulation for both schools in HT and LT areas.

### 2.7. Comet Assay

Buccal epithelial cells of children were collected for the analysis of DNA damage using Comet assay. The assay was conducted following the protocol in Comet assay Kit (Trevigen, Gaithersburg, MD, USA; catalogue no. 4253-096-K). Firstly, the buccal cells suspension were washed with 1x PBS, calcium and magnesium free (Thermo Fisher Scientific, Waltham, MA, USA; catalogue no. 70011-044) and centrifuged at 2500 rpm for 1 min. 10 µL of cells were combined with 75 µL of low melting agarose and immediately pipetted onto the comet slide (Trevigen, Gaithersburg, MD, USA; catalogue no. 4250-200-03). The slides were placed flat in the dark at a 4° C chiller. After 30 min, the slides were immersed in a pre-chilled lysis solution (Trevigen, Gaithersburg, MD, USA; catalogue no. 4250-050-01) and left for 60 min in 4 °C chiller. Next, the excess buffer was discarded from the slides. The slides were immersed in the freshly prepared alkaline unwinding solution for 60 min at room temperature. The slides were removed from the alkaline solution and were placed horizontal onto the electrophoresis slide tray. Then, alkaline electrophoresis solution was poured in the tray until the level covered over the slides. Electrophoresis was run for 20 min at 21 V constant voltage. The slides were washed twice in deionized water for 5 min and then washed once in 70% ethanol (R&M Chemicals, Birmingham, UK; CAS no. 64-17-5) for 10 min at room temperature. After drying, the slides were stained with 50 µL SYBR Green (Invitrogen, Carlsbad, CA, USA; catalogue no. S7563). The comet images were captured using fluorescence microscope (Olympus, Tokyo, Japan; model BX51) under 20× magnification. The images were analysed using OpenComet software version 1.3 (https://www.cometbio.org, accessed on 24 September 2021). The extent of DNA damage was reported as tail moment (the product of %T and tail length).

### 2.8. Statistical Analysis

The data collected were analysed using Statistical Package for Social Science (SPSS) Version 25 (IBM, Armonk, NY, USA). The data were checked for missing values and outliers. Data normality of continuous variables were determined based on Shapiro-Wilks. The statistical tests include univariate, bivariate, and multivariate analysis to prove the relationship between PAHs and DNA damage. Univariate analysis included descriptive statistics for sociodemographic data, PAHs concentration, and tail moment (descriptor for DNA damage). Bivariate analysis involved comparison of means or medians of the variables studied, particularly for continuous data. The mean for tail moment was log10-transformed to get normal-distributed data prior to statistical analysis. Meanwhile, multiple linear regression was performed to assess the main variables that influence the DNA damage among the respondents.

### 2.9. Quality Control

The pre-test of the questionnaire was conducted on at least 10% of the total respondents, which was among 28 respondents. The Cronbach’s α value was 0.812 which considered acceptable (α ≥ 0.7). The buccal samples were kept at 4 °C in an icebox with ice packs until being transported to laboratory. The comet cells were examined in a zig-zag pattern under fluorescence microscope to prevent over counting or repetition of cells. Filter papers were pre-baked at 500 °C for 4 h using muffle furnace (Carbolite, Sheffield, UK; model CWF11/23). To avoid photodegradation, the filter papers were wrapped up in aluminium foil in sealed bags. No plasticware was used for PAHs analysis to avoid any cross-contamination from other sources. All glassware used during the procedure were cleaned, rinsed with n-hexane and rinsed with distilled water and acetone (R&M Chemicals, Birmingham, UK) prior to being baked in a furnace at 500 °C for 4 h, to volatilize and remove any organic contaminants.

## 3. Results and Discussion

### 3.1. Distributions of PAHs Species at Schools

Figure 2 shows the concentrations of total PAHs (∑PAHs) in indoor and outdoor environment at each school. The distributions of PAHs species in indoor and outdoor PM_2.5_ samples were shown in Appendix A. As portrayed in Figure 2, the trend of indoor ∑PAHs concentration for HT schools are as follows H1 (5.58 ± 4.72 ng m^−3^) > H3 (4.86 ± 3.17 ng m^−3^) > H4 (4.65 ± 1.39 ng m^−3^) > H2 (4.19 ± 0.91 ng m^−3^). Meanwhile for LT schools, L2 demonstrated the highest indoor ∑PAHs concentration (3.69 ± 3.19 ng m^−3^) followed by L1 (3.48 ± 3.14 ng m^−3^), L4 (2.26 ± 1.32 ng m^−3^) and L3 (1.25 ± 0.83 ng m^−3^). The results clearly showed that a higher concentration of outdoor ∑PAHs was seen in HT schools compared to LT schools with the highest concentration recorded at school H1 (5.76 ± 2.20 ng m^−3^), closely followed by school H3 (5.69 ± 3.22 ng m^−3^), school H4 (4.96 ± 1.96 ng m^−3^) and school H2 (4.40 ± 1.79 ng m^−3^). On the other hand, the schools in LT group portrayed low PAHs concentration, especially school L3 (1.36 ± 0.69 ng m^−3^) and L4 (2.63 ± 1.96 ng m^−3^). H1 is located less than 50 m away from the main road and 160 m from highways. Meanwhile, L3 recorded the lowest ∑PAHs concentrations, with 1.25 ± 0.83 ng m^−3^ for indoor and 1.36 ± 0.69 ng m^−3^ for outdoor. The low ∑PAHs obtained were probably due to less traffic densities in that area. Moreover, the school environment surrounded by forest may act as a natural filter that contributes to removing air pollutants, thus lowering the concentration of total ∑PAHs [39]. Trees can help minimize air pollution by absorbing the particulate matter via stoma uptake [40,41].

In this study, BkF was the dominant species presence in particulate PAHs samples for all HT schools with concentration range from 0.64 to 1.84 ng m^−3^ for indoor and 0.71 to 1.04 ng m^−3^ for outdoor. The result is consistent with study by Suradi et al. [18] which detected BkF out of 13 measured PAHs species, as the dominant species at Kuala Lumpur City Hall (DBKL) with the mean concentration 0.42 ng m^−3^. On the other hand, ACP was the highest species found in the indoor samples of school L1 and L2 with the concentration of 0.91 ng m^−3^ and 0.84 ng m^−3^, respectively.

It was found that the range of the concentration of indoor ∑PAHs obtained in this study was quite similar with study by Ismail et al. [39] conducted in three primary schools in Kuala Lumpur, Malaysia with the concentration ranged from 1.6 to 8.0 ng m^−3^ for indoor PAHs. BgP was the dominant species found in the study, which indicate the vehicular emission. Another study conducted by Sopian et al. [42] in school environment in Terengganu, Malaysia reported a higher concentration than the present study with concentration ranged from 4.21 to 63.22 ng m^−3^ and 5.93 to 67.72 ng m^−3^ for indoor and outdoor, respectively. Furthermore, the present study reported a much lower PAHs concentration than other studies dealing with the school environment in China [37,43], Portugal [26] and Lithuania [44].

An urban middle school in Beijing China recorded 6 times higher total PAHs concentration than the present study with the concentration of 29.8 ng m^−3^ and 33.7 ng m^−3^ for indoor and outdoor, respectively [37]. They had identified BbF, CYR, IcP, Flu, BgP and BaP as the most prominent species and might be emitted from coal combustion, residential cooking and traffic exhausts. Wang et al. [43] found a higher level of indoor and outdoor PAHs in school located in commercial and residential area of Xian, China with concentrations of 79.9 and 92.00 ng m^−3^, respectively. BbF, IcP, BgP, CYR and BkF were the most abundant species found in that study, which contributed by biomass burning, vehicle emissions, and coal combustion activity. PAHs exposure in primary urban schools in Portugal recorded a higher total PAHs concentration range from 2.8 to 54 ng m^−3^ for indoor and 7.1 to 48 ng m^−3^ for outdoor which are highly dominated by DhA, Acy and BjF species [26]. Meanwhile, Krugly et al. [44] reported the concentrations of particulate and gaseous PAHs in indoor and outdoor air from five Lithuanian urban primary schools with concentration range from 20.3 to 131.1 ng m^−3^ and 40.7 to 121.2 ng m^−3^ for indoor and outdoor, respectively. In that study, naphthalene appeared be the most abundant PAH species in all sampling sites. Motor vehicle emissions and fuel combustion for heating purposes were the main sources of PAHs in the study.

### 3.2. Indoor and Outdoor PAHs

Figure 3 shows the indoor to outdoor (I/O) ratio of PAHs concentrations. The I/O ratio can be used to explain the relationship between indoor and outdoor pollution states [37]. If the I/O ratio were >1, indoor sources are stronger than outdoor sources. On the other hand, if it were <1, the indoor sources are weaker [45]. This study shows that all schools had I/O ratios below 1, ranging from 0.86 to 0.99, indicating that the indoor sources were relatively weak. The results of I/O ratio show that among eight schools, school L2 recorded the highest ratio (0.99). This situation can be explained by a higher penetration of outdoor particles into indoor classrooms resulting in higher I/O ratios at L2. This phenomenon may be due to the indoor and outdoor air exchange due to the natural ventilation system, which refers to the opening of windows and doors in this school. In addition, the windows in school L2 were open most of the time, which increases the airflow into the classrooms. The pathway of pollutants from outdoor air to indoor spaces depends on ventilation and infiltration. Prevailing wind provides natural ventilation when doors and windows are open [46]. Infiltration can also occur through cracks and leaks in the building, which can be significant for a poorly sealed building [47]. Because of these mechanisms, outdoor air pollutants can enter indoor spaces and can either be diluted or accumulated depending on ventilation conditions [47].

Previous work has conducted studies on the I/O ratio in school environments. Wang et al. [43] reviewed more than 16 studies from different schools around the world and found that the I/O ratio of PAHs ranged from 0.43 to 0.93. A study in a middle school in Beijing, China reported almost similar I/O ratio with the present study, with a value of 0.98 [37]. According to Long and Sarnat [48], an indoor source is present when the I/O ratio is greater than 1.15. The interpretation of the I/O ratios in the present study supports the idea that indoor PAH concentrations are mainly influenced by the ambient atmosphere. Analysis of the diagnostic ratios of individual PAHs may provide further insight into the origin of PAHs.

### 3.3. Distribution of PAHs Based on Number of Rings

The percentage distribution of outdoor and indoor PAHs based on number of rings was shown in Appendix A. In general, the distribution of individual PAHs was mainly dominated by high molecular weight (HMW) PAHs structured by four to six rings (FLT, PYR, BaA, CYR, BaP, BbF, BkF, BgP, DhA, IcP), compared to low molecular weight (LMW) PAHs with three rings (ACY, ACP, FLR, PHE, ANT). However, school L1, L2, and L4 demonstrated a contra finding as the LMW conquered the total PAHs concentration compared to HMW. The indoor and outdoor PAHs at school L1, L2 and L4 had higher contribution of three aromatic ring PAHs, accounting for more than 50% of the total measured PAHs. The 4-ring PAHs in all schools accounted for 10 to 20% of the total PAHs. The outdoor HT school, H3 had the highest fraction of 4 ring (20%). The 5 and 6 ring PAHs were vastly abundant (ranges between 51 to 79%) in all schools except for three schools L1, L2 and L4. The three mentioned schools had 32.62% (L1), 27.24% (L2) and 37.68% (L3) of HMW PAHs for the outdoor samples. Furthermore, the indoor samples of 5 and 6 ring PAHs were highly detected in school H3 with a percentage of 78.88% of the total measured PAHs.

In urban areas, pyrogenic sources are the main source of PAHs, especially HMW-PAHs, which are mainly present in PM_2.5_. [18]. As suggested by Yunker et al. [49], HMW PAHs are more likely to be portioned in PM_2.5_ compared to LMW PAHs. Based on the finding, a higher distribution of HMW PAHs was seen in all HT schools which consistent with previous studies in urban traffic areas in the Klang Valley [10,11,19,36,50]. HMW PAHs that consist of more than 5 ring are an indicator of traffic emission [51]. HMW PAHs are usually formed during the process involving high temperature such as fuel combustion [39]. Meanwhile, the formation of LMW PAHs is associated with low-temperature combustion, such as wood-burning [52].

### 3.4. Source Diagnostic Ratio

The source diagnostic ratio was used to identify possible PAH sources in each school. The application of diagnostic ratios involves comparing ratio between specific pairs of PAHs compounds with the same molar mass and similar physicochemical properties [52]. Bivariate plots of selected PAHs based on source diagnostic ratio are shown in Figure 4. The ratio ANT/(ANT + PHE) was indicative of an anthropogenic source of PAHs emissions, with values below 0.1 indicating a petrogenic source. In contrast, any value above 0.1 indicates a pyrogenic source [53]. In this study, the ratio ANT/(ANT + PHE) had a value greater than 0.1, indicating a strong contribution from the pyrogenic source. In addition, the ratio of IcP/(IcP + BgP) was greater than 0.2 for all samples, indicating a contribution from a pyrogenic source such as fossil fuel combustion, grass, wood, or coal.

The BaA/(BaA + CYR) ratio is reported as an indicator for petrogenic (unburned petroleum) sources when <0.2 and combustion sources when >0.35. A ratio between 0.2 and 0.35 indicates that it is from mixed sources [49]. Most of the samples had the ratios of BaA/(BaA + CYR) higher than 0.35 suggesting that the combustion activity is a primary source. Akyüz and Çabuk [54] also suggested that a ratio greater than 0.35 denotes vehicular emissions. Two important anthropogenic sources, namely gasoline and diesel emissions, contribute to traffic-related air pollution. The FLR/(FLR + PYR) ratio can identify diesel and gasoline vehicle sources with the higher ratio (>0.5) of FLR/(FLR + PYR) indicate a diesel emission. In contrast, a lower ratio (<0.5) indicates a gasoline emission [55,56]. Majority of schools in this study were impacted by diesel emission, except for H3 and L3. Interestingly, 100% of samples from H2 and H4 were originated from diesel emission, which could be contributed by the pass-by of heavy-duty vehicles. Meanwhile, L3 had the highest number of samples (n = 6, 75%) originated from the exhausts of gasoline engine vehicles. This study’s findings suggested that PAH sources mainly from vehicular emission and other pyrogenic contributions such as grass, wood, and coal combustion. A similar finding was found in a previous study that reported PAHs compound in Kuala Lumpur may not only originated from urban traffic combustion, but also contributed by coal, grass, and wood burning activities [18].

### 3.5. Health Risk Assessment

The obtained values for ΣTEQ–PAHs in outdoor air ranged from 0.67 to 2.77 ng m^−3^ with an overall average of 1.64 ng m^−3^ (Appendix A). Meanwhile, the obtained values for ΣTEQ–PAHs in indoor air ranged from 0.86 to 2.87 ng m^−3^ with an overall average of 1.62 ng m^−3^ (Appendix A). Generally, the TEQ values reported in this study, except for school L3 and L4, has exceeded the maximum permissible risk level of 1 ng m^−3^ of BaP as set by European Guidelines. Oliveira et al. [28] reported that TEQ values were mainly higher in Asian schools (range: 4.70–49.4 ng m^−3^) compared to European school environments (range: 0.04–29.8 ng m^−3^).

DhA congener exhibited the highest carcinogenic potency of the PAHs in all schools, and this was most likely due its high toxicity factor (TEF value of 5). On average, DhA contributed up to 80% and 76% of outdoor and indoor ΣTEQ–PAHs, respectively. In contrast, BaP, the most known and studied carcinogenic PAHs, was the second-highest contributor for ΣTEQ–PAHs, accounting for 13.5% at outdoor and 17.5% at indoor air. The result is consistent with several previous studies [43,57,58], which reported DhA and BaP as the most influential components in the TEQ values.

The children in the HT group had a significantly higher risk of cancer than the children in the LT group. It can be arranged in ascending order: L3 (5.41 × 10^−8^) < L4 (6.18 × 10^−8^) < L2 (9.39 × 10^−8^) < L1 (9.43 × 10^−8^) < H2 (1.04 × 10^−7^) < H4 (1.27 × 10^−7^) < H3 (1.66 × 10^−7^) < H1 (1.99 × 10^−7^). Monte Carlo simulations were performed to make an accurate cancer risk estimation of PAH inhalation, as shown in Figure 5. The mean value of the probability cancer risk for the HT and LT groups were 1.54 × 10^−7^ and 7.59 × 10^−8^, respectively, and both means show similarity to the actual ILCR generated in the SPSS software (1.59 × 10^−7^ and 7.58 × 10^−8^). The 95th percentiles of the ILCR calculated the risk for the HT and LT populations were 2.80 × 10^−7^ and 1.43 × 10^−7^, respectively. These ILCR values indicate that the daily inhalation dose of PAHs and cancer risk for children in the HT group is lower than the acceptable levels of 10^−6^ to 10^−4^ as proposed by the USEPA (2011). The estimated carcinogen risk for the HT group was higher than previous studies in Kuala Lumpur with a calculated ILCR of 2.64 × 10^−8^ and 5.51 × 10^−8^ in 2019 and 2021, respectively [11,18].

A sensitivity analysis was performed to find the parameter with the greatest impact on the overall risk outcome. In this study, the TEQ value and body weight (BW) were the most influential parameters for the carcinogenic risk due to PAH inhalation. As shown in Figure 6, the PAHs concentration in the TEQ value has a greater impact on the ILCR estimation in HT and LT group, with a correlation coefficient of 0.58 and 0.71, respectively. In contrast, an inverse relationship appeared between BW and estimated carcinogenic risk for HT and LT group (correlation coefficient: −0.72 and −0.59) which similarly found by several previous studies [42,59,60,61]. A negative correlation value indicates that the increase in predictor is associated with a decrease in ILCR prediction. The results of the sensitivity analysis suggest that the values and probability distributions of TEQ and BW should be correctly determined to increase the accuracy of the estimated risks.

### 3.6. Individual Factors on DNA Damage

The result shows that the tail moment (parameter of DNA damage) of the HT group (3.13 ± 0.53) were significantly higher than the value recorded in the LT group (2.80 ± 0.81) with *p* < 0.05. The individual factors on tail moment were stratified and compared using independent t-test and one-way ANOVA, as shown in Table 2. Considering all children together, the present study revealed that children aged 10–11 years old had significantly higher tail moment compared to children aged 7 to 9 years old (3.03 ± 0.62 vs. 2.84 ± 0.71). Several studies have reported an increased DNA damage by older age as assessed by the comet assay [62,63]; meanwhile there are studies that showed no effect of age on the extent of DNA damage [64,65].

Gender, age, and BMI are among the important demographic parameters that need to be emphasized in epidemiological studies because of their influence on genotoxic effects [66]. In this study, no significant difference in the tail moment were found for gender. Gajski et al. [65], in a study of healthy children living in an urban area in Croatia, found a significantly higher mean value of comet assay parameters in female children compared to male children. In the present study, no significant difference in tail moment measurement was found for BMI categories, which is in agreement with study by Sopian et al. [42]. It is well documented that being overweight and obese are associated with an increased DNA damage [67]. Gandhi [68] reported that DNA damage was almost five folds higher in the young obese subjects when DNA migration (strand breaks) was compared to the healthy control. The study speculated that increased oxidative stress and depletion of antioxidant in obese subjects resulted in increased genetic damage.

The effects of tobacco smoking on DNA damage have been widely investigated because tobacco contains carcinogenic and genotoxic substances. Therefore, smoking is also considered as a confounder [69]. In the present study, children who exposed to environmental tobacco smoke (ETS) had a higher tail moment (3.01 ± 0.57) than those children who did not exposed to ETS (2.95 ± 0.70); however, no significant difference detected from the mean comparison. Studies by Zalata et al. [70] and Beyoglu et al. [71] demonstrated that exposure to ETS among children was positively proven can increase DNA damage. School-aged children are particularly more exposed to secondhand smoke compared with other age groups. School-aged children spend a lot of time at home and stay close to their parents, suggesting that living with parents who smoke may be a strong predictor of increased exposure to substances contained in cigarettes [72].

Interestingly, the results showed that children living less than 500 m from main road and highway had a longer tail moment than the children living more than 500 m from main road and highway. However, the differences were statistically significant only for house distance from highway (*p* = 0.028) but not with house distance from main road (*p* = 0.465). Meanwhile, mode of transportation to school, did not exhibit any significant difference in tail moment measurement. In this study, grilled food consumers had a higher tail moment than children who consume less frequently (3.03 ± 0.61 vs. 2.96 ± 0.67, *p* = 0.554). Children who less frequently consume supplement also were observed had a slightly higher tail moment (2.99 ± 0.70) than children who frequently take supplement (2.96 ± 0.61). Fruit consumption, however, showed inverse findings, as children who frequently eat fruit had a higher tail moment than those who less frequently eat fruit (3.00 ± 0.65 vs. 2.81 ± 0.66, *p* = 0.123). High consumption of vegetables, fruits and juices rich in antioxidant vitamins and phytophenols has been shown to be positively associated with low levels of endogenous DNA strand breaks and oxidised DNA bases, and protective against ex vivo generation of DNA damage [69,73].

### 3.7. Factors That Influence DNA Damage

Simple linear regression was applied to investigate the relationship between DNA damage (tail moment) with possible predictor factors of DNA damage among children. Tail moment was used as a dependent variable in the linear regression. Appendix A shows that the total particulate PAHs and carcinogenic PAHs in indoor and outdoor environment had significant influence on the occurrence of DNA damage in buccal epithelial cells of children in this study. In addition, the analysis also revealed a significant association between house distance from highway and DNA damage. Other individual factors did not exhibit significant association in the regression model.

The significant variables were further computed into MLR analysis using a stepwise method to determine the best predictor of the dependent variable. It was revealed that total indoor PAH exposure was the most significant factor that influenced the DNA damage among children, as portrayed in equation below:(3)$$Log\;tail\;moment=2.734+0.063\;\left(Exposure\;to\;total\;indoor\;PAHs\right)$$

As shown in Table 3, the model predicted every unit increase of exposure to total indoor PAHs will lead to increment of log tail moment by 0.063. For the model, the beta value was significant at the 0.05 level. VIF value was <5, which suggested that there was no problem with multicollinearity. There was a significant direct linear relationship between total indoor PAHs with DNA damage (*p* < 0.05). Total indoor PAHs explained 4.4% of the variance in tail moment, adjusted *R^2^* = 0.044, F (1, 226) = 11.54, *p* < 0.05.

The present study strongly suggests that DNA damage is significantly affected by indoor particulate PAHs after controlling all possible confounding factors such as age, gender, BMI, house distance from main road and highway, mode of transportation to school, dietary habit and ETS exposure. Since all classrooms in this study has natural ventilation system, the present study provide evidence that children could be subjected to outdoor PAHs that infiltrate the schools’ indoor environment via open windows and doors, in which poses a higher risk of genotoxicity. In addition, the indoor-outdoor PAHs ratio in all schools ranged from 0.86 to 0.99, indicating a high penetration of outdoor PAHs into indoor classrooms.

The present study is in line with those of Sopian et al. [42], who discovered severe DNA damage in children living near a petrochemical plant in Terengganu, which exposed them to high amounts of particulate PAHs. In that study, the total indoor PAHs and open burning were the significant factor that influence the tail moment (adjusted *R^2^* = 0.127). In other words, the interaction of indoor PAH emissions and open burning significantly explained a 12.7% variation of the tail moment. In another study by Jasso-Pineda et al. [27], a significant DNA damage was found in a group of children living in a family that utilised biomass combustion. The study revealed a significant correlation between internal PAHs exposure (1-OHP) and DNA damage (r = 0.65, *p* < 0.01). Similarly, Sanchez Guerra et al. [74] also found positive associations between internal PAHs exposure (1-OHP) and DNA damage among Mexican children living near petrochemical industries. In addition, the results of this study are supported by a study by Ismail et al. [75], which demonstrated a high risk of DNA damage and respiratory symptoms in children who attended school near busy roads in Selangor.

A fact that could be considered a limitation of the present study was other toxic pollutants emitted from motor vehicles were not investigated in this study. DNA damage could also be induced by numerous environmental pollutants besides PAHs (i.e., Benzene and 1,3-butadiene) [76,77]. Future research should be conducted on quantification of other carcinogenic pollutants such as toxic gases (benzene, toluene, ethylbenzene, xylenes) to evaluate comprehensively the combine effects of exposure to urban air pollutants on genotoxic effects in children.

## 4. Conclusions

The findings provided evidence that children living near busy roads are more likely to be exposed to environmental PAHs and have a higher risk of genotoxicity than children living in low traffic areas. This study has successfully reduced the knowledge gap on relationship between urban traffic pollution and genotoxicity in children in the Southeast Asia region. The results not only contribute to the understanding of the levels, distribution, and sources of PAHs in educational settings, but also shed light on the governance of children’s living environments and well-being, especially in urban areas. Although damage to genetic material is ubiquitous and inevitable for organisms, effective DNA repair systems can protect against these negative effects and maintain genetic stability. However, if the rate of DNA damage exceeds the capacity of the cell to repair it, detrimental biological consequences such as genotoxic damage and carcinogenesis will occur. Further study on the link between PAH exposure and genomic integrity in children should be investigated to shed additional light on potential health risks. This study recommends increasing the distance between future school sites and busy roads to reduce children’s exposure to PAHs from traffic. Although the location of schools should be further away from busy roads, it is not easy to relocate existing schools, especially in the Klang Valley where the population is growing but there is less and less land available. Therefore, traffic density around existing schools should be reduced.

## Figures and Tables

**Figure 1 ijerph-19-02193-f001:**
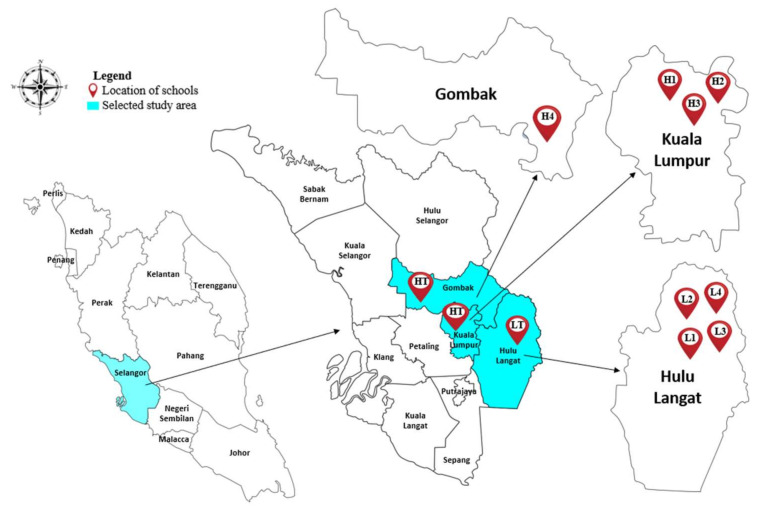
Locations of the selected primary schools. Primary schools in Kuala Lumpur and Gombak, were categorised as high traffic (HT) group with the schools in HT were designated H1, H2, H3 and H4. Primary schools in Hulu Langat, were categorised as the low traffic (LT) group, with the schools on LT designated as L1, L2, L3, and L4.

**Figure 2 ijerph-19-02193-f002:**
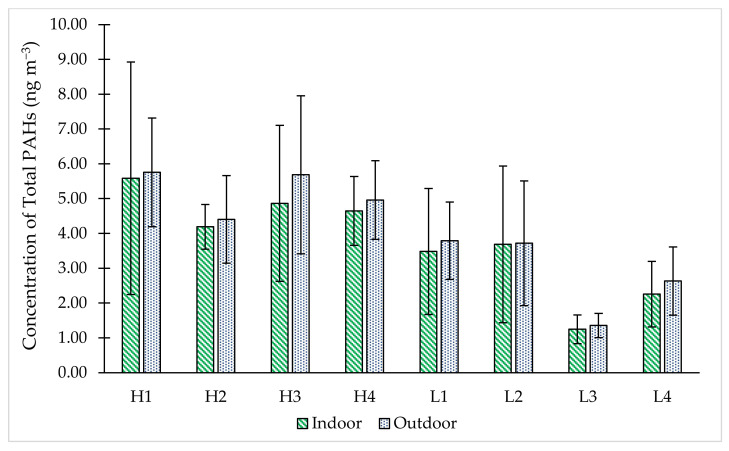
The concentrations of total PAHs in indoor and outdoor environment at each school.

**Figure 3 ijerph-19-02193-f003:**
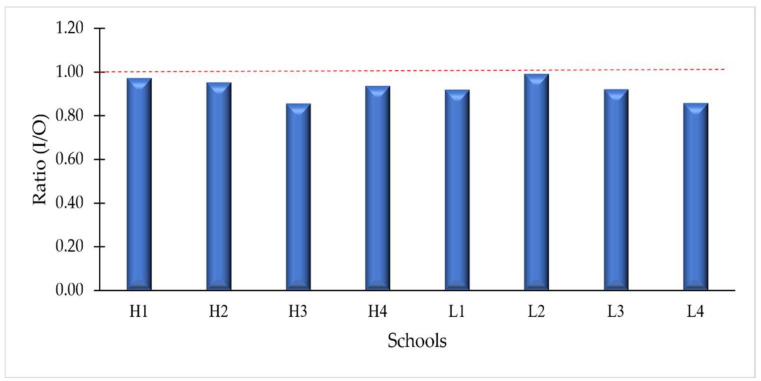
Indoor to outdoor (I/O) ratios of PAHs concentration in each school.

**Figure 4 ijerph-19-02193-f004:**
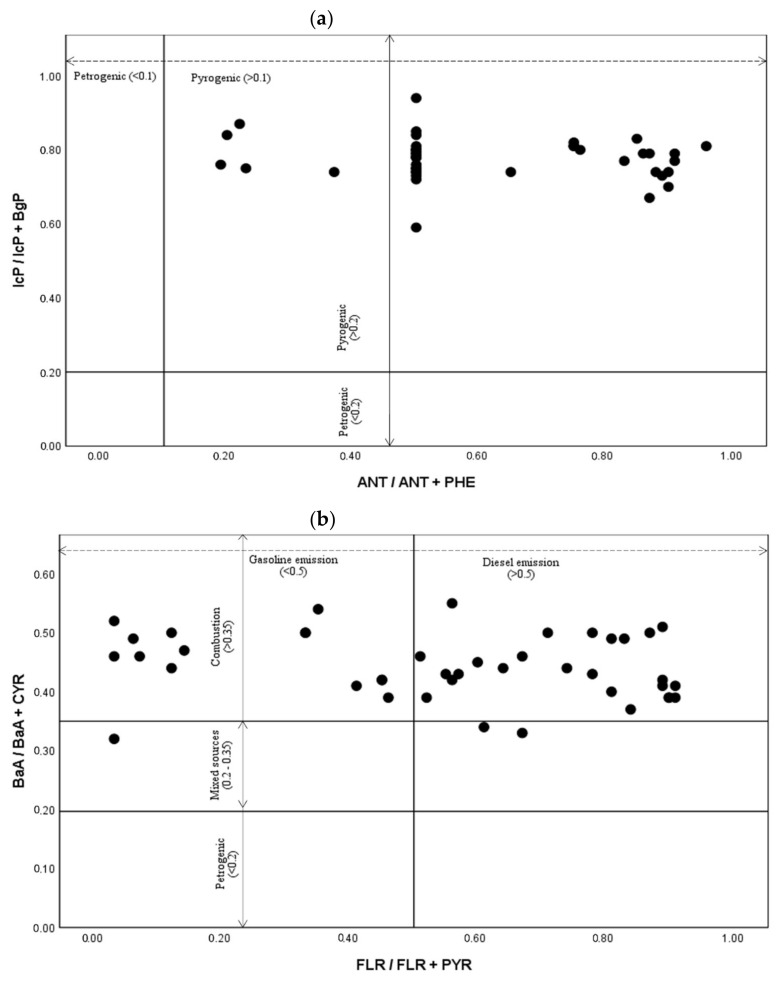
Result of source diagnostic ratio method. (**a**) IcP/IcP + BgP and ANT/ANT + PHE (**b**) BaA/BaA + CYR and FLR/FLR + PYR.

**Figure 5 ijerph-19-02193-f005:**
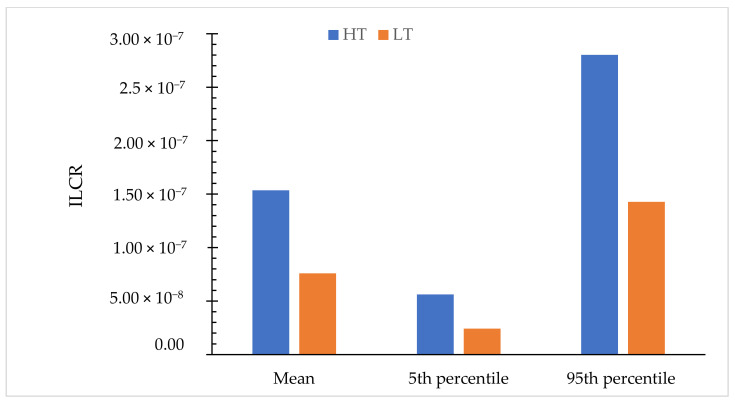
Probabilistic distribution of ILCR for children in HT and LT groups.

**Figure 6 ijerph-19-02193-f006:**
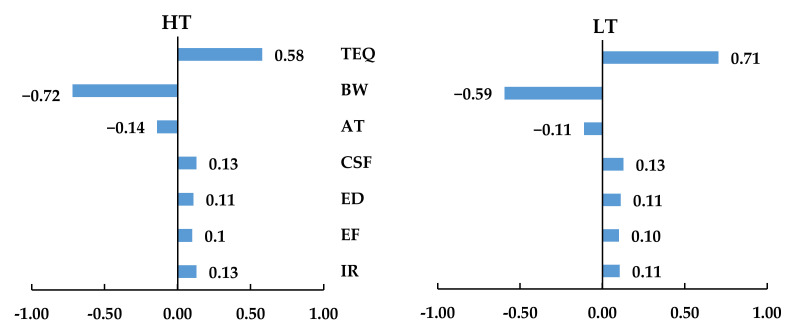
Sensitivity analysis of carcinogenic risk for children in HT and LT groups. TEQ = Toxic equivalent concentration; BW = Body weight; AT = Averaging time of carcinogenic PAHs exposure; CSF = Inhalation cancer slope factor; ED = exposure duration; EF = Exposure frequency; IR = Inhalation rate.

**Table 1 ijerph-19-02193-t001:** The variable distribution types applied in the Monte Carlo simulation for both schools in HT and LT areas.

Parameter	Distribution Mode	HT Schools	LT Schools
TEQ (ng m^−3^)	Logistic *	2.32	1.02
Inhalation rate (m^3^ day^−1^)	Constant	12	12
Exposure frequency (day year^−1^)	Constant	250	250
Exposure duration (year)	Constant	6	6
Averaging time (days)	Constant	25, 500	25, 500
Body weight (kg)	Log normal *	33.13, 14.54	28.33, 10.86
Cancer slope factor (mg kg^−1^ day^−1^)	Constant	3.85	3.85

* Logistic data are represented as arithmetic mean, log normal represented as LN (arithmetic mean and standard deviation).

**Table 2 ijerph-19-02193-t002:** Comparison of Tail Moment according to a different individual factor.

Variables	HT (N = 113)		LT (N = 115)		All Children (N = 228)	
	GM ± SD	*p-*Value ^‡^	GM ± SD	*p-*Value ^‡^	GM ± SD	*p-*Value ^‡^
Age						
7–9	3.15 ± 0.50	0.914	2.74 ± 0.74	0.302	2.84 ± 0.71	0.040 *
10–11	3.13 ± 0.54		2.88 ± 0.71		3.03 ± 0.62	
Gender						
Boy	3.09 ± 0.53	0.310	2.86 ± 0.66	0.556	2.99 ± 0.60	0.773
Girl	3.19 ± 0.54		2.78 ± 0.77		2.96 ± 0.71	
BMI categories						
Underweight	3.20 ± 0.59	0.338	2.98 ± 0.62	0.421	3.10 ± 0.60	0.138
Normal	3.19 ± 0.45		2.87 ± 0.64		3.02 ± 0.58	
Overweight	2.84 ± 0.61		2.53 ± 1.13		2.68 ± 0.88	
Obese	3.07 ± 0.63		2.69 ± 0.85		2.88 ± 0.76	
House distance from main road						
<500 m	3.13 ± 0.53	0.846	2.85 ± 0.71	0.243	2.99 ± 0.64	0.465
≥500 m	3.16 ± 0.58		2.63 ± 0.82		2.90 ± 0.75	
House distance from highway						
<500 m	3.16 ± 0.49	0.504	2.59 ± 0.44	0.393	3.11 ± 0.51	0.028 *
≥500 m	3.09 ± 0.59		2.83 ± 0.74		2.91 ± 0.71	
Mode of transportation to school						
Active mode	3.04 ± 0.59	0.509	2.79 ± 0.98	0.928	2.95 ± 0.73	0.899
Motorized mode	3.15 ± 0.53		2.82 ± 0.71		2.98 ± 0.65	
Grilled food						
Yes	3.14 ± 0.54	0.990	2.94 ± 0.66	0.387	3.03 ± 0.61	0.554
No	3.13 ± 0.54		2.79 ± 0.74		2.96 ± 0.67	
Supplement intake						
Yes	3.12 ± 0.49	0.750	2.76 ± 0.69	0.466	2.96 ± 0.61	0.763
No	3.15 ± 0.59		2.86 ± 0.75		2.99 ± 0.70	
Fruit consumption						
Yes	3.14 ± 0.54	0.617	2.85 ± 0.72	0.274	3.00 ± 0.65	0.123
No	3.06 ± 0.47		2.66 ± 0.73		2.81 ± 0.66	
ETS exposure						
Yes	3.16 ± 0.52	0.672	2.87 ± 0.58	0.516	3.01 ± 0.57	0.495
No	3.12 ± 0.55		2.78 ± 0.80		2.95 ± 0.70	

^‡^ ANOVA and Independent Sample *t*-test were computed based on log-transformed DNA damage; * Significant at *p* < 0.05.

**Table 3 ijerph-19-02193-t003:** Factor that influences DNA damage among children after controlling all confounders.

Variable	*B* (95% CI)	β	*p*-Value	Adjusted R^2^
Constant	2.734 (2.572, 2.897)		<0.001 *	0.044
Indoor tPAHs	0.063 (0.026, 0.100)	0.220	0.001 *	

* Significant at *p* < 0.05; Method: Stepwise.

## Data Availability

Not applicable.

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
