# Peer review of "The Influence of Environmental Polycyclic Aromatic Hydrocarbons (PAHs) Exposure on DNA Damage among School Children in Urban Traffic Area, Malaysia"

_ijerph, 2022, doi:10.3390/ijerph19042193_

Round 1
Reviewer 1 Report
See the attached document .

Reviewer 2 Report
This study is valuable in the sense that it provides data in support of the correlation between cancer risk exposure in children and environmental PAH contamination level. The findings of the study are non-surprising and rather expected. A closer proximity of schools to heavy-traffic areas (i.e. higher levels of PAHs released into the environment) reflects an increased health concern.
Some things to keep in mind when revising the manuscript are:
- Line 145: Six-point calibration seems to have been performed (rather than five point as mentioned)
- Figures 2a and 2b: Please add error bars in order for readers to be able to better appreciate magnitude of differences.
- There is no "s" after PAH when used as attribute of a noun, i.e. for example in "PAH samples" (lines 14 and 114); PAH concentrations (line 18); PAH exposure (lines 24, 68, 463, and 525).
- Line 34: "humans" (plural)
- Line 106: rephrase to "and 228 respondents completed..."
- Line 229: rephrase to "Results clearly showed that..."
- Line 254: rephrase to "other studies dealing with the school environment..."
Reviewer 3 Report
Dear authors:
The paper is well-writing and shows interesting results, although some improvements could be included:
- Like PAH and UFP, prolonged exposure to volatile organic compounds (VOCs; e.g. BTEX compounds) also has been related to cell damage. VOCs are usually detected in relatively high concentration (mg/m3) in urban areas [e.g. 10.1016/j.chemosphere.2020.128127]. So, this study could be considered limited or biased by not including VOCs. Please justify
- It is suggested that the results and findings of this work be discussed and contrasted with similar studies found in the literature.
- All images look fuzzy. Consider the use of high-definition images.
